# A Minimal Genetic Passkey to Unlock Many Legume Doors to Root Nodulation by Rhizobia

**DOI:** 10.3390/genes11050521

**Published:** 2020-05-07

**Authors:** Jovelyn Unay, Xavier Perret

**Affiliations:** Microbiology Unit, Department of Botany and Plant Biology, Sciences III, University of Geneva, 30 quai Ernest-Ansermet, CH-1211 Geneva 4, Switzerland; Jovelyn.Unay@unige.ch

**Keywords:** legume–rhizobia symbioses, symbiotic promiscuity, synthetic replicon, plant immune responses

## Abstract

In legume crops, formation of developmentally mature nodules is a prerequisite for efficient nitrogen fixation by populations of rhizobial bacteroids established inside nodule cells. Development of root nodules, and concomitant microbial colonization of plant cells, are constrained by sets of recognition signals exchanged by infecting rhizobia and their legume hosts, with much of the specificity of symbiotic interactions being determined by the flavonoid cocktails released by legume roots and the strain-specific nodulation factors (NFs) secreted by rhizobia. Hence, much of *Sinorhizobium fredii* strain NGR234 symbiotic promiscuity was thought to stem from a family of >80 structurally diverse NFs and associated nodulation keys in the form of secreted effector proteins and rhamnose-rich surface polysaccharides. Here, we show instead that a mini-symbiotic plasmid (pMiniSym2) carrying only the *nodABCIJ*, *nodS* and *nodD1* genes of NGR234 conferred promiscuous nodulation to ANU265, a derivative strain cured of the large symbiotic plasmid pNGR234a. The ANU265::pMiniSym2 transconjugant triggered nodulation responses on 12 of the 22 legumes we tested. On roots of *Macroptilium atropurpureum*, *Leucaena leucocephala* and *Vigna unguiculata*, ANU265::pMiniSym2 formed mature-like nodule and successfully infected nodule cells. While cowpea and siratro responded to nodule colonization with defense responses that eventually eliminated bacteria, *L. leucocephala* formed leghemoglobin-containing mature-like nodules inside which the pMiniSym2 transconjugant established persistent intracellular colonies. These data show seven nodulation genes of NGR234 suffice to trigger nodule formation on roots of many hosts and to establish chronic infections in *Leucaena* cells.

## 1. Introduction

Unlike many angiosperms, including cereals, legume crops can form beneficial nitrogen-fixing associations with soil bacteria called rhizobia. Provided legume cultivars and rhizobia strains are optimally matched, these symbioses can substitute for nitrogen fertilizers, thus lessening the ecological impacts of agriculture [1]. In most legumes, reduction of atmospheric nitrogen (N_2_) is restricted to root nodules inside which rhizobia establish persistent intracellular colonies of N_2_-fixing bacteroids [2]. Infection of nodules cells is generally mediated by infection threads (ITs), which form at the tip of curled root hairs and guide rhizobia across several cortical cell layers towards the developing nodule primordia, where ITs branch to contact a greater number of newly formed nodule cells [3,4]. Eventually, infecting rhizobia are released from ITs and colonize the cytoplasm of nodule cells in the form of symbiosome compartments made of one or several bacteroids enclosed within a peribacteroid membrane of plant origin [5]. As infected nodule cells may each contain several hundreds of bacteroids, a fine-tuning of the plant immune system must allow rhizobia colonization of nodules while prevent a systemic infection of roots or the entry of pathogenic microorganisms [6,7]. Most legume crops form one of two main nodule types. Nodules with indeterminate growth meristem (indeterminate nodules, IDN) possess a persistent apical meristem that maintains a gradient of differentiating plant and rhizobia cells across the nodule longitudinal section [8,9]. Plants such as alfalfa (*Medicago sativa*), chickpea (*Cicer arietinum* L.) and the tree *Leucaena leucocephala* form IDN that, once mature, have elongated shapes. By contrast, determinate nodules (DN) are spherical, possess a transient meristem, differentiate in a more synchronous manner and eventually contain populations of bacteroids often considered as developmentally homogenous across a nodule section. Cowpea (*Vigna unguiculata*), siratro (*Macroptilium atropurpureum*) and soybean (*Glycine max*) are examples of legumes crops with DN. With time, some DN may also develop secondary clusters of dividing cells that form new nodule meristems, as was shown for *Tephrosia vogelii* (and by analogy *Cajanus cajan*) [10]. The developmental and infection processes leading to the formation of proficient nodules where bacteroids can fix nitrogen are complex, and sets of molecular cues exchanged by plants and rhizobia coordinate both of these processes [11,12]. In such a molecular dialog, signals emitted by one of two partners and the corresponding cognate receptors of the second symbiont were proposed to function together as key-lock systems that define the specificity of rhizobia–legume associations [13,14,15].

Cocktails of root flavonoids and of rhizobia nodulation (Nod) factors (NF) contribute greatly to secure beneficial symbioses [2,12,16]. In rhizobia, compatible flavonoids and LysR-type regulators of the NodD family act together to positively regulate the transcription of genes found downstream of conserved promoter sequences called *nod* boxes (NB) [16,17,18]. Most of NB-controlled operons code for enzymes responsible for the synthesis of Nod factors made of β-1,4-linked *N*-acetyl-d-glucosamine (GlcNAc) subunits with a fatty acyl group attached to the non-reducing saccharide [19]. NF synthesis requires a chitin oligosaccharide synthase NodC; one N-deacetylase NodB; and the acyl transferase NodA. The number of GlcNAc residues, type of fatty acyl moiety and additional chemical substitutions (e.g., acetylation, methylation, sulfation, etc.) make NF strain specific [16,20]. Once synthesized, NF are secreted through the inner bacterial membrane via the NodI and NodJ proteins [21,22,23], with rhizobia often secreting more than one kind of NF. In principle, micro- to nanomolar concentrations of purified or synthetic NF suffice to initiate nodule development on roots of compatible hosts [24], and to allow rhizobia entry into host roots [25,26]. Legumes perceive NF via membrane bound LysM receptor kinases (LRK) that bind compatible NF with high affinity [27] and whose roles in symbiosis have been reviewed recently [28,29]. Together NF and cognate receptors are thought to ensure symbiotic specificity, as altered NF structures changed the host-range of mutant rhizobia [19,30,31,32] and swapping of LysM domains between hosts modified NF-perception and specificity to rhizobia [33]. Rhizobia may also use type three or type four secretion systems (T3SS or T4SS) to secrete effector proteins that modulate symbiotic interactions [34,35], or help bypass NF-dependent nodulation pathways as in some soybean cultivars [36], or in *Aeschynomene indica* [37]. While NF are important signal molecules for triggering nodule formation and securing entry of rhizobia into root hairs, bacterial cell surface components (CSC) such as exo- (EPS) and lipo-polysaccharides (LPS) were shown to play critical roles during ITs development [11]. For example, perception of compatible EPS by the EPR3 LRK system of *Lotus japonicus* secures the elongation of ITs across root cell layers [38], with *Mesorhizobium loti* EPS mutants being blocked inside aborted ITs and consequently unlikely to reach nodules [39]. Thus, infection of root nodules by rhizobia appears as gated by a number of successive checkpoints (doors) that rhizobia must pass using specific molecular signals (keys) (e.g., NF, CSC, secreted effectors) aimed at the corresponding cognate plant receptors (e.g., NF- and EPS-specific LRK complexes) positioned along the infection corridor. Although number and types of clearing steps may differ between legumes, they contribute together to limit the entry of nodule cells to mostly genuine microsymbionts.

Such a complex framework of molecular interactions befitted well the specificity of many rhizobia–legume symbioses that, almost a century ago, prompted researchers to define cross-inoculation groups of proficient symbionts [40]. With a host-range of >120 legume genera and its capacity to reduce N_2_ inside DN or IDN of >150 plant species [41], the fast growing *Sinorhizobium* (*Ensifer*) *fredii* strain NGR234 appeared as both, an exception to the specificity rule and a model to study the genetics of symbiotic promiscuity [42]. Except for a few chromosomal genes [43,44], most of the symbiotic genes required for nodulation (*nod*, *nol* and *noe*) and nitrogen fixation (*nif* and *fix*) are coded by the 536 kb pNGR234a plasmid [45]. Hence, the ANU265 derivative strain cured of pNGR234a was unable to form nodules (Nod- phenotype) [46]. In NGR234, the flavonoid-dependent expression of >75 pNGR234a genes involved in nodulation is controlled by a complex NodD1-TtsI-SyrM2-NodD2 regulatory cascade [47,48]. In this regulatory network, NodD1 is activated by a broad spectrum of flavonoids and acts as a master switch for symbiosis [49,50]. The NodD1-dependent cascade controls numerous factors that were proposed to contribute to NGR234 promiscuity [13], including the secretion of a pool of >80 structurally different NF [51], the T3SS-dependent translocation of five known effectors [34,35,52] and the synthesis of rhamnose-rich modified LPS [53]. Apparently, such a large “key ring” of symbiotic cues made NGR234 more promiscuous than rhizobia equipped with fewer nodulation “keys”.

Here, we show instead that pMiniSym2, an artificial and low-copy mini-symbiotic plasmid (pMiniSym) carrying only the *nodABCIJS* and *nodD1* genes of pNGR234a allowed ANU265 to trigger the nodule developmental program on roots of 9 of 14 legume species which associate with NGR234. Depending on the host, the nodulation phenotype of the ANU265::pMiniSym2 transconjugant ranged from empty pseudonodules to infected mature-like nodules that expressed leghemoglobin. Intermediate phenotypes of plants responding to nodule infection with signs of enhanced immune responses were also observed. These findings indicate that a skeleton NF key of NGR234 already unlocks the doors to many legume roots, but, when nodules are formed, does not necessarily secure a sustained infection of these nodule structures.

## 2. Materials and Methods

### 2.1. Bacterial Strains, Plasmids and Growth Conditions

Bacterial strains and plasmids are listed in Appendix A. *Escherichia coli* recombinants were grown at 37 °C in/on Luria-Bertani medium [54]. *S. fredii* strains NGR234, ANU265 and derived transconjugants were grown at 27 °C in/on tryptone yeast (TY) medium [55] and *Rhizobium* minimal medium (RMM) supplemented with 12 mM succinate (RMS) as sole carbon source [56]. When needed, rifampicin, spectinomycin and kanamycin were added at concentrations of 50 µg/mL.

### 2.2. Assembly of Mini Symbiotic Plasmids

Prior to assembly of mini symbiotic plasmids (pMiniSym), the necessary gene blocks were amplified separately and cloned as individual DNA fragments in pBluescript II KS+ (Stratagene, La Jolla, CA, USA). Genes or operons of interest, as well as primers and DNA templates used to amplify them with the proof reading PrimeStar^®^ HS DNA polymerase (Takara Bio, Inc., Kusatsu, Shiga, Japan) are listed in Appendix A. Amplified products were cloned as separate *Spe*I- or *Kpn*I-products into pBluescript II KS+ (Stratagene, La Jolla, CA, USA) and the sequence of cloned amplicons was verified by Sanger sequencing (Microsynth, Balgach, Switzerland). Prior to assembly with the Gibson Assembly protocol [57], each of the selected gene blocks was amplified with the PrimeStar^®^ HS DNA polymerase using the corresponding pBluescript clone as DNA template and a set of specific overlapping primers (see Appendix A). Gene blocks for pMiniSym1 (*ori*V, *ori*T, *trfA*, *spsAB* and Km^R^ Omega interposon) were assembled together for 1 h at 50 °C using the Gibson assembly^TM^ mix (New England Biolabs, Cambridge, UK), with aliquots of the assembly mix transformed into *E. coli* DH5α. For pMiniSym2, the nodulation gene blocks (for *nodABCIJ*, *nodS* and *nodD1*) were first assembled together for 15 min at 50 °C prior to be added onto linearized pMiniSym1 amplicons using NEBuilder^®^ Hifi DNA assembly mix (New England Biolabs, Cambridge, UK) for 1 h at 50 °C. Assembly products for pMiniSym2 were transformed into *E. coli* strain NEB^®^10-β (New England Biolabs, Cambridge, UK) and sequence verified before being conjugated into ANU265 by triparental mating using pRK2013 as helper plasmid [58]. A derivative pMiniSym2-Gus construct expressing β-glucuronidase constitutively was obtained by cloning the *uidA* gene of *E. coli* under the control of the *rpsL* promoter of NGR234 [59] as a 2.6 kb *Spe*I fragment (see Appendix A for primers and template). Plasmid sequences of pMiniSym1, pMiniSym2 and pMiniSym2-Gus are deposited in GenBank under accession numbers MN266286, MN266287 and MN266288, respectively.

### 2.3. Plant Assays and Analysis of Root Nodules

All nodulation assays were carried out in Magenta jars containing sterile vermiculite, nitrogen-free B&D solution [60] and germinated seedlings having been surface sterilized as previously described [59]. Two days after planting in vermiculite, each plantlet was inoculated with a 200 μL water suspension of 2 × 10^8^ freshly grown rhizobia using a minimum of 6 plants per inoculum and per experiment. After several weeks of growth at a day temperature of 27 °C, a night temperature of 20 °C and a light phase of 12 h with 60–70% humidity, plants were harvested, and roots were examined for nodule or pseudonodule formation. Nodule bacteria were isolated as previously described [61] and their identity confirmed by PCR and DNA sequencing. For Gus staining, hand-sectioned nodules or whole root systems were vacuum infiltrated in X-Gluc buffer [0.1% sarcosyl, 0.1% triton X100, 5 mM K-ferricyanide, 5 mM K-ferrocyanide, 1 mM EDTA and 0.2-μg/mL X-Gluc (5-bromo-4-chloro-3-β-glucuronic acid)] using a Red Evac^TM^ (Hoefer Scientific Instruments, San Francisco, CA, USA) and incubated at 37 °C for 12 to 48 h, depending on signal intensity [59]. Nodule sections were observed using a Leica MZ16 equipped with an Infinity 2 CCD camera (Leica Microsystems, Wetzlar, Germany). To obtain micrographs, nodules of *V. unguiculata* and *M. atropurpuruem* were vacuum infiltrated at RT for 2 h in a 4% (v/v) paraformaldehyde and 5% (v/v) glutaraldehyde in 0.1 M sodium-cacodylate buffer (pH 7.2) fixative solution and post-fixed at 4 °C for 2 h in a 1.5% (w/v) osmium tetroxide (OsO_4_) solution. After another 1 h treatment at 4 °C in 1% (w/v) uranyl acetate, nodule sections were dehydrated in a series of ethanol solutions and embedded in Epon resin (Sigma-Aldrich, St. Louis, MO, USA). Ultra-thin (85 nm) nodule sections were obtained with a Leica UCT ultramicrotome (Leica Microsystems, Wetzlar, Germany), stained with 2% (w/v) uranyl acetate and Reynolds’ lead citrate solution and observed using a TECNAI G2 Sphera (FEI Company, Hillsboro, OR, USA) transmission electron microscope at 120 kV. For light microscopy, semi-thin sections (1 μm) were stained with 75-µg/mL methylene blue for 1 min at 65 °C followed with 0.5 mg/mL basic fuchsine for 5 min at room temperature. Once stained, sections were viewed under Nikon Eclipse 80i microscope (Nikon, Shinagawa, Tokyo, Japan). For nodules of *L. leucocephala*, samples were fixed ON in 4% paraformaldehyde and 2.5% glutaraldehyde in 0.1-M sodium-cacodylate buffer (pH 7.0), post-fixed for 1 h under vacuum in 2% (w/v) osmium tetroxide, dehydrated in acetone series and embedded in Spurr resin (Electron Microscopy Sciences, Hatfield, PA, USA). Semi-thin sections (0.5 μm) were obtained using a Reichert microtome (Leica Microsystems, Wetzlar, Germany) and stained with 75-µg/mL methylene blue for 3 min at 65 °C followed with 0.5 mg/mL basic fuchsine for 5 min at RT and observed under a Nikon Eclipse 80i microscope (Nikon, Shinagawa, Tokyo, Japan).

### 2.4. Detection of Plant Immune Responses

Accumulation of phenolic compounds was detected by staining nodule sections with potassium permanganate (KMNO_4_) as described previously [62,63]. Briefly, freshly sliced agarose-embedded nodule sections (300 μm) were post-fixed in 0.02% (w/v) KMNO_4_ and 5 mM PIPES [piperazine-N,N′-bis (2-ethanesulfonic acid); pH 7.2] for 45 min at RT. Once rinsed with 10 mM PIPES, sections were stained with 0.01% methylene blue for 10 min and then cleared in 4% (v/v) sodium hypochlorite (NaOCl) until unspecific stain was removed. Once rinsed with water, sections were photographed using Leica MZ16 equipped with an Infinity 2 CCD camera (Leica Microsystems, Wetzlar, Germany).

### 2.5. qPCR-Based Determination of pMiniSym2 Copy Number

Cells of ANU265::pMiniSym2 and NGR234 were grown in TY and RMS containing the appropriate antibiotic until OD_600_~0.8 and total genomic DNA (gDNA) was extracted using phenol-chloroform [54]. Concentration of purified gDNA was determined with a NanoDrop One (Thermo Fischer Scientific, Waltham, MA, USA). Quantitative polymerase chain reaction (qPCR) was carried out in a QuantStudio^TM^ 5 thermal cycler (Applied Biosystems, Foster City, CA, USA) using PowerUp SYBR^®^ Green Master Mix (Thermo Fischer Scientific, Waltham, MA, USA), 10 μL reaction volumes, concentrations of 0.156, 0.6, 2.5, 10 and 40 ng gDNA as template and primers for the plasmid-borne *nodB* and *nodD1* genes and the chromosome-borne *rpoC* and *rpsL* loci (see Appendix A). Plasmid copy number (PCN) was obtained using the 2[Ct(Chr)-Ct(plas)] formula where Ct(Chr) for chromosome and Ct(plas) for plasmid correspond to averaged Ct’s for chromosomal *rpoC* and *rpsL* genes and plasmid *nodB* and *nodD1* genes, respectively [64].

## 3. Results

### 3.1. Design and Assembly of pMiniSym1, pMiniSym2 and pMiniSym2-Gus Replicons

In rhizobia, plasmids carrying nodulation and nitrogen fixation genes are mostly of the RepABC family and are low to unit-copy number plasmids [65]. Accordingly, pMiniSym1, the genetic chassis for mini-symbiotic replicons included the origin of replication (*oriV*), trans-acting replication factor A (*trfA*) gene and origin of transfer (*oriT*) from the low-copy number and broad-host-range plasmid pRK7813 of the RK2 family [66], as well as the post-segregational killing system SpsAB of pNGR234a [67]. The 7.2 kb long pMiniSym1 also carried a kanamycin resistant (Km^R^) Omega interposon [68] to facilitate selection of transformed or transconjugant cells and to transcriptionally insulate the other genetic modules [59,68] (see Figure 1). To obtain pMiniSym2, several genes of the nodulation regulon of NGR234 were then added to pMiniSym1, namely: the *nodD1* gene under the control of its native promoter, to secure a flavonoid-dependent expression of nodulation genes [49,50]; the *nod*-box 8 (NB8) and NodD1-controlled *nodABCIJ* operon that is required for the synthesis and secretion of pentameric NF [51]; and the NB12- and NodD1-controlled *nodS* gene for N-methylation of NF [69], thereby also contributing to sufficient NF levels [13]. As shown in Figure 1 the *nodABCIJ*, *nodS* and *nodD1* genetic modules of pMiniSym2 were organized and oriented such as to minimize interference during flavonoid-dependent transcription. Subsequently, the β-glucuronidase gene (*uidA*) under the control of the *rpsL* promoter of NGR234 (P*rpsL*) was cloned into the unique *Spe*I restriction site of pMiniSym2. Since P*rpsL*-dependent expression was shown to be strong in free-living and symbiotic conditions [59,70], this additional feature helped tracking pMiniSym2-Gus transconjugant bacteria on synthetic media or on roots, as well as inside nodules. Except for the *Spe*I-mediated cloning of the P*rpsL*-*uidA* cassette, assembly of pMiniSym plasmids was carried out in vitro using the Gibson Assembly protocol [57].

### 3.2. pMiniSym2 Is Maintained as a Low-Copy Replicon in ANU265

To determine the plasmid copy number (PCN) of pMiniSym constructs, we compared by quantitative polymerase chain reaction (qPCR) the relative numbers of single-copy and housekeeping *rpoC* and *rpsL* genes of the NGR234 chromosome and of the symbiotic *nodD1* and *nodB* genes carried by both, pNGR234a and pMiniSym2. Total genomic DNA (gDNA) of NGR234 and ANU265::pMiniSym2 strains was purified from cells growing exponentially in RMS (minimal) or TY (rich) media. Depending on the concentration of gDNA used as template (ranging from 0.156 to 40 ng in 5 incremental steps), the PCN of pNGR234a was established at 1.2 to 1.5 copies and 1.1 to 1.4 copies, when cells of NGR234 were grown in RMS and TY, respectively (see Appendix A). By contrast, in transconjugant cells of ANU265, the PCN of pMiniSym2 was estimated at 1 to 1.3 copies and 1.8 to 2.1 copies, when cells were grown in RMS and TY, respectively. This indicated that, unlike for the pNGR234a PCN that remained remarkably stable at ca. 1 to 1.5 copy independently of the growth medium, PCN of pMiniSym2 increased slightly when cells of ANU265 were grown in TY. Nevertheless, pMiniSym2 PCN remained close to the unit-copy number of symbiotic plasmids found in rhizobia.

### 3.3. pMiniSym2 Confers Nodulation to ANU265 Transconjugants

Once pMiniSym2 was mobilized into ANU265, the nodulation properties of the resulting transconjugant (ANU265::pMiniSym2) were tested on 22 different legumes belonging to 19 species that are listed in Table 1. All plants were grown in Magenta^TM^ jars and under controlled conditions using as control for nodulation strain NGR234 or, when NGR234 was either a poor or inefficient symbiont on the targeted legume, another proficient symbiont (e.g., *Mesorhizobium japonicum* strain MAFF303099 for Lotus species). Given such an experimental setup, four weeks of incubation was always sufficient for NGR234 or the alternative control strain to induce nodule formation on roots of compatible hosts (Nod+ phenotype) and 8 of the 22 legumes tested already showed improved growth when challenged with NGR234. As examples, shoots of *M. atropurpureum* cv. Siratro and *V. unguiculata* cv. Red Caloona that were inoculated with NGR234 had greener leaves and were more developed than those of non-inoculated control plants, which showed signs of nitrogen starvation (see Appendix A). When harvested, non-inoculated control plants and those challenged with ANU265 never formed nodules or other root structures indicative of a nodule developmental response (Nod- phenotype). By contrast, ANU265::pMiniSym2 triggered a robust nodulation response on all roots of 5 different legumes as shown in Appendix A for *M. atropurpureum* cv. Siratro, *L. leucocephala* and *V. unguiculata* cv. Red Caloona. Interestingly, while the ANU265::pMiniSym2 transconjugant formed mature-like nodules albeit devoid of leghemoglobin on siratro and cowpea, those triggered on roots of *Leucaena* were pink (Figure 2 and Appendix A). On *Flemingia congesta* and *Lablab purpureus*, response to ANU265::pMiniSym2 inoculation was irregular (Nod+/−) with some roots failing to display any visible nodule development at 28 dpi. Some legumes, such as *Vigna radiata* cv. King and two *C. cajan* cultivars, plants responded to inoculation with ANU265::pMiniSym2 by formation of small pseudonodules or bumps (pNod), however. *Crotalaria juncea*, *Desmodium intortum*, *Lotus japonicus* cv. Gifu and *Glycine canescens* 007, all of which were nodulated by NGR234, failed to exhibit any visible nodulation response (Nod-) when challenged with ANU265::pMiniSym2. On *Pachyrhizus tuberosus*, onto which NGR234 irregularly forms few nodules (Nod+/−) whereas a T3SS-mutant becomes fully proficient [52], no discernible nodule structure was triggered by ANU265::pMiniSym2. Similarly, the pMiniSym2-transconjugant failed to elicit a detectable response on the roots of plants that are non-host of NGR234 such as *Cicer arietinum*, *Lotus corniculatus* cv. Malejovsky, *M. sativa* cv. Gemini, *M. truncatula* cv. Jemalong and *Mimosa pudica* [41].

### 3.4. On Several Legume Hosts, pMiniSym2 Allows ANU265 to Reach Nodule Cells

Isolation of viable bacteria from 15 to 21 dpi surface sterilized nodules of *M. atropurpureum* cv. Siratro, *L. leucocephala* and *V. unguiculata* cv. Red Caloona indicated that ANU265::pMiniSym2 strain was capable of reaching nodule tissues. To better follow the path of rhizobia infection, we used as inoculum the ANU265::pMiniSym2-Gus and NGR234::pXPrpsL426 [59] derivative strains that both constitutively express the *β*-glucuronidase (Gus) reporter enzyme. On *V. unguiculata* and *M. atropurpureum*, 28 dpi nodule sections showed the ANU265::pMiniSym2-Gus transconjugant failed to establish chronic infections, as only small patches of plant tissues displayed detectable levels of *β*-glucuronidase activity (see Appendix A). By contrast, nodules triggered by NGR234::pXPrpsL426 transconjugant were fully stained, as expected of a strain that establishes proficient symbioses on both cowpea and siratro (Appendix A). Interestingly, on *L. leucocephala*, both NGR234::pXPrpsL426 and ANU265::pMiniSym2-Gus formed fully stained nodules, indicating the ANU265 transconjugant could extensively colonize those IDN and retain pMiniSym2-Gus, even in the absence of antibiotic pressure, throughout the numerous division cycles needed to fully infect *L. leucocephala* nodules (Appendix A).

### 3.5. ANU265::pMiniSym2 Establishes Intracellular Colonies in L. leucocephala Nodule Cells

The β-glucuronidase activity detected throughout the infected zone of *L. leucocephala* nodules suggested that ANU265::pMiniSym2-Gus strain had chronically infected nodule tissues. To determine whether that infection was intracellular, as in genuine symbiotic associations, nodules at different stages of development were examined by light and electron microscopy. As shown in Figure 3, plant cells of 21 dpi nodules contained many NGR234 bacteroids (Figure 3a,b), had small vacuoles and electron-dense cytoplasm, with intracellular bacteria enclosed within symbiosomes delimited by peribacteroid membranes (Figure 3b,d). By contrast, nodule cells infected with ANU265::pMiniSym2 harbored fewer bacteria (Figure 3e,f), which were dispersed inside cellular compartments much larger than regular symbiosomes (Figure 3f,g). Interestingly, the cytoplasm of both NGR234 and ANU265::pMiniSym2 often contained poly-*β*-hydroxybutyrate (PHB) granules (Figure 3b–d,f,g) suggesting intracellular bacteria had access to ample carbon sources even though nodules formed by the ANU265 transconjugant were not fixing nitrogen. Thus, micrographs of nodule sections confirmed that ANU265::pMiniSym2 successfully established persistent intracellular colonies inside *L. leucocephala* nodules, although bacteria were not enclosed in normal symbiosome structures.

### 3.6. Infection of Cowpea and Siratro Nodules by ANU265::pMiniSym2 is Countered by Plant Immune Responses

Given the ANU265 cells retained pMiniSym2-Gus for several weeks inside *L. leucocephala* nodules, plasmid instability did not seem a likely reason for the limited and patchy Gus staining that was observed in *M. atropurpureum* and *V. unguiculata* nodules at 28 dpi. As necrotic lesions were frequently seen in sections of cowpea and siratro nodules formed by ANU265::pMiniSym2 (see Appendix A), both legumes appeared as defending against infections by the transconjugant. To test whether phenolic compounds, which are often associated to plant immune responses [72], could be detected inside nodules formed by ANU265::pMiniSym2, the nodule sections were treated with potassium permanganate and then stained with methylene blue. As shown in Appendix A, sections of nodules formed by NGR234 on roots of *M. atropurpureum* and *V. unguiculata* were not stained whereas those infected by ANU265::pMiniSym2 were stained blue, indicating presence of phenolic compounds.

To verify whether bacteria had nonetheless gained access to nodule cells, we examined by light and electron microscopy cowpea and siratro nodules that were harvested at the earlier time point of 15 dpi. As shown in Appendix A, cowpea nodules formed by NGR234::pXPrpsL426 and ANU265::pMiniSym2-Gus had similar ultra-structures, with a central infected zone made of large infected plant cells separated by smaller non-infected interstitial cells and with vascular bundles surrounding the infected zone. Electron micrographs confirmed that in the case of plants inoculated with NGR234::pXPrpsL426, the cytoplasm of infected nodule cells contained large numbers of bacteroids (Appendix A) that were enclosed in symbiosomes delimited by peribacteroid membranes (Appendix A). Interstitial nodule cells often contained large starch granules (Appendix A) and infection of nodule cells by NGR234::pXPrpsL426 was mediated via ITs as shown in Appendix A. By contrast, in 15 dpi cowpea nodules formed by ANU265::pMiniSym2-Gus, the infected nodule cells had lost a normal compartmentalization and contained fewer intracellular bacteria (Appendix A), many of which appeared as being actively degraded (Appendix A). Infection of cowpea nodule cells by ANU265::pMiniSym2-Gus possibly also occurred via ITs, although no such normal structures were observed. Instead, abnormal ITs or infection pockets containing degenerated bacteria were observed, suggesting defense reactions against infecting bacteria (e.g. Appendix A). As shown in Appendix A, the ANU265::pMiniSym2-Gus transconjugant suffered a similar fate inside nodules of *M. atropurpureum* with degradation of bacteria occurring in both disintegrating nodule cells as well as in necrotic lesions (Appendix A). By contrast, numerous NGR234::pXPrpsL426 cells had established functional symbiosomes inside plant cells of 15 dpi siratro nodules (Appendix A). Together, these observations confirmed that ANU265::pMiniSym2 had colonized nodule cells of *V. unguiculata* and *M. atropurpureum*, but plant defenses prevented the transconjugant from establishing persistent colonies inside of these nodule cells.

## 4. Discussion

Symbiotic nitrogen fixation requires rhizobia to establish persistent intracellular colonies inside nodules, where host plants provide the sheltered environment that allow populations of symbiotic bacteria to expend. Yet, it is only once intracellular rhizobia have differentiated into proficient bacteroids, that root nodules become a source of fixed N and do not remain a sink for plant resources. Seedlings must thus carefully balance the energy reserves at their disposal, before investing the resources needed to develop nodules and the bacteroid populations that inhabit them. To avoid being lured into non profitable associations, as well as being infected by soil pathogens, legumes have evolved complex mechanisms to guide rhizobia towards nodule primordia, for example via infection threads, while screening these infecting bacteria for compatible symbionts (for reviews see [2,73,74]).

For many, if not most rhizobia, the synthesis of appropriate NF is essential to initiate symbiosis and failure to produce adequate NF levels and/or appropriate NF structures make these strains unable to form nodules [2,11,16,75]. Perception of NF by cognate legume receptors was repeatedly documented as a key step in the establishment of symbiotic interactions, thus contributing extensively to the specificity of nodulation [28,29]. Thus, the complex cocktail of >80 structurally different NF synthesized by the promiscuous strain NGR234 befitted well the key–lock model in which one kind of NF (one key) opened the door to one or a few legume hosts, providing they shared receptors with similar NF affinities [13]. Here, we showed that the situation is more complex, since the ANU265::pMiniSym2 transconjugant that lacks functional copies of the *nodU*, *nodZ*, *noeE*, *noeI*, *noeJ*, *noeK*, *noeL*, *nolK*, *nolL* and *nolO* genes of NGR234, still gained access to many legume doors. Response to inoculation with ANU265::pMiniSym2 varied greatly between legumes that are hosts of NGR234, from no visible response (documented as Nod- in Table 1) to mature-like nodules expressing leghemoglobins (*L. leucocephala*), indicating the minimum set of nodulation keys needed for a sustained nodule developmental program differs considerably between these plants. That ANU265::pMiniSym2 failed to induce visible nodule primordia on *C. juncea*, *D. intortum* and *G. canescens* (on which NGR234 forms nodules) may have resulted from a combination of factors, among which missing host-specific NF decorations is one of the possibilities. Modified levels of secreted NF when compared to NGR234 may also contribute to define the nodulation properties of ANU265::pMiniSym2. Although PCN of pMiniSym2 in ANU265 was shown to be comparable to that of pNGR234a in NGR234, we cannot exclude that levels of NF secreted by the two strains are significantly different and not necessarily with the pMiniSym2 transconjugant strain synthesizing higher NF levels than NGR234. While mass action of non-specific NF may possibly trigger a nodule developmental program on some hosts, excess amount of NF was conversely shown to be detrimental to optimal infection and nodule development in *M. truncatula* [76]. T3SS-dependent effectors may also modulate nodulation responses and absence of a functional T3SS is likely to have contributed to the reduced nodulation capacity of ANU265::pMiniSym2 on several hosts, including *T. vogelii* (pNod) that was shown to require T3SS effectors for optimal nodulation by NGR234 [77]. In several rhizobia-legume combinations, T3SS effectors were shown to either promote [78] or abort nodule development [79] and in the case of incompatible interactions mediated by rhizobia T3SS, nodule primordia were shown to stop their development at an early stage [80]. Taken together, these considerations indicate that our failure to detect visible nodule structures on roots of several of the NGR234 hosts did not mean these plants were necessarily blind to the NF secreted by ANU265::pMiniSym2.

In the absence of a reporter tool that would facilitate scoring of plant responses, even subtle, at the onset of nodulation and on all of the tested legume species, we concentrated instead on legumes that responded to inoculation by ANU265::pMiniSym2 with the formation of mature-like nodules. In particular, we analyzed the late stages of the infection processes on *M. atropurpureum*, *L. leucocephala* and *V. unguiculata*. Cowpea and siratro belong to the same *Phaseoleae* tribe and both make DN while *L. leucocephala* belongs to *Mimoseae* tribe and makes IDN. Using the ANU265::pMiniSym2-Gus strain that expresses the β-glucuronidase constitutively, we showed that bacteria colonized cowpea nodules harvested at 14 dpi, but failed to establish persistent colonies in older nodules (e.g., at 28 dpi; Appendix A). Electron micrographs of 15 dpi nodule sections confirmed the presence of bacteria within intercellular infection pockets between (Appendix A) and inside nodule cells (Appendix A), but those few intracellular bacteria failed to form symbiosomes and seemed to be targeted for degradation (Appendix A). In fact, plant defense responses detected in the form of phenolic compounds and necrotic plant cells must have contributed to eliminate nodule bacteria over time as we found it increasingly difficult to isolate live bacteria from cowpea nodules older than 21 dpi. These data showed plants actively suppressed live bacteria weeks before nodules begun to decay. Plant defense responses are not necessarily limited to incompatible rhizobia–legume associations, since necrotic plant cells were observed on alfalfa seedlings close to the site of *Sinorhizobium meliloti* infection [63], however. Like on cowpea, the ANU265::pMiniSym2 transconjugant triggered mature-like nodules on siratro roots and micrographs of nodule sections confirmed nodule cells to be intracellularly infected, but again poorly when compared to those housing NGR234 (Appendix A). Siratro nodules seemed to remain infected longer than cowpea nodules, which was consistent with the fewer necrotic lesions and milder accumulation of phenolic compounds detected in siratro nodules (Appendix A). Interestingly, when compared to the mature-like nodules formed by cowpea when challenged with ANU265::pMiniSym2, a closely related species such as *V. radiata* only responded with smaller pseudonodules. Given *V. unguiculata* and *V. radiata* not only share taxonomic proximity, but must also often coexist in fields of tropical areas, these distinct nodulation patterns indicate that NF perception systems and downstream plant signaling cascades may evolve rapidly to tailor host-specificity of nodulation to meet the plant’s requisite. In this respect, that a strain carrying a nodulation regulon smaller than the narrow host-range *Cupriavidus taiwanensis* strain LMG19424 [81] still induces a nodule developmental program on many legumes, including hosts as taxonomically distant as cowpea (Papilionoideae) and *L. leucocephala* (Caesalpinioideae), challenges our current understanding on NF and their role in defining symbiotic promiscuity. In principle, the limited set of nodulation genes (*nodD1*, *nodABCS* and *nodIJ)* carried by ANU265::pMiniSym2, suffice for the synthesis and the secretion of NF devoid of most of the “baroque decorations” [16] found on NGR234 NF (see Appendix A). In a seminal publication [82], Madsen and associates showed NF triggered in roots of *L. japonicus* two parallel processes: DN organogenesis and infection thread formation. Sustained infection of roots by rhizobia was also shown to be required for a continued nodule development in IDN forming plants such as *Pisum sativum* [83] and *M. truncatula* [84]. Given the absence of specific NF-decorations was shown to be detrimental to entry of rhizobia into specific hosts, but not for root hair deformation and nodule primordium development on these same hosts [85], it is conceivable that some of the decorations missing on ANU265::pMiniSym2 NF may be required for bacterial entry and thus sustained nodule development on the legume hosts that failed to form mature-like nodules. Although, more detailed analyses of the early stages of the infection processes will be needed to confirm such a hypothesis, the construction of pMiniSym replicons carrying additional nodulation genes offers a way for systematic analysis of the role(s) of specific NF decorations on a broad range of legumes.

Unlike the few mature-like, but poorly infected nodules found on cowpea and siratro roots, *L. leucocephala* responded to inoculation of pMiniSym2 and pMiniSym2-Gus transconjugants of ANU265 with nodules that were similar in size, color and numbers to those made by NGR234 (see Appendix A), including for the presence of leghemoglobin (Figure 2, panels g1 to g2*) and persistent intracellular colonies of bacteria (Figure 3). Leghemoglobin is often considered as a marker for active symbiotic nitrogen fixation. As the ANU265 transconjugant lacked any of the nitrogen fixation genes, the presence of leghemoglobin throughout nodules suggests that *Leucaena* initially perceived the nodulating bacterium as a genuine symbiont or either failed to detect it as a non-fixer. Nodule development also persisted over several weeks, since plants harvested at 49 dpi with ANU265::pMiniSym2 carried three-to-four-mm-long root nodules that were not pink anymore, but still contained live bacteria (data not shown). Such a peculiar nodulation phenotype sustained over at least seven weeks—in complete absence of a source of fixed N, and in association with a pseudo-rhizobia strain incapable of nitrogen fixation and that lacks all of the other known symbiotic hallmarks of NGR234—makes *Leucaena* a host more tolerant to infection than *V. unguiculata* or any other legume tested so far. Obviously, experimental conditions where host legumes are grown in absence of reduced N and in contact with a single rhizobia strain are unlike any of the natural growth conditions encountered by legumes in fields. Yet, the remarkable association between *L. leucocephala* and ANU265 containing pMiniSym replicons provides opportunities to test more complex synthetic symbiotic plasmids in which minimal sets of nitrogen fixation genes will be regulated by NifA.

## Figures and Tables

**Figure 1 genes-11-00521-f001:**
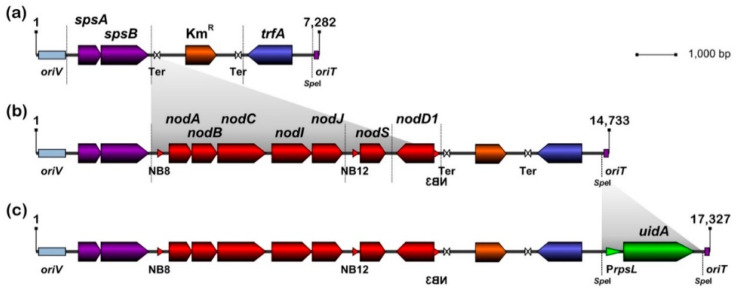
Genetic maps of pMiniSym1, pMiniSym2 and pMiniSym2-Gus. In the linear genetic maps of pMiniSym1 (**a**), pMiniSym2 (**b**) and pMiniSym2-Gus (**c**), each of the assembled genetic modules is delimited by vertical dashed lines. pMiniSym1 carries functions for propagation (TrfA, *oriV*), selection (Km^R^ Omega interposon) and maintenance (*spsAB*) in *E. coli* and rhizobia, as well as a conjugative origin of transfer (*oriT*). In addition to pMiniSym1 functions, pMiniSym2 includes a small set of nodulation genes that confers to recipient bacteria, perception of plant flavonoids by NodD1, synthesis of pentameric methylated nodulation factor (NF) (by NodA, NodB, NodC and NodS) and their secretion (by NodI and NodJ), via a NodD1-dependent transcriptional activation mediated by the NB8 and NB12 nod boxes. pMiniSym2-Gus carries the *uidA* reporter gene for β-glucuronidase (Gus), which constitutive expression is controlled by the *rpsL* promoter (P*rpsL*) of NGR234 [59].

**Figure 2 genes-11-00521-f002:**
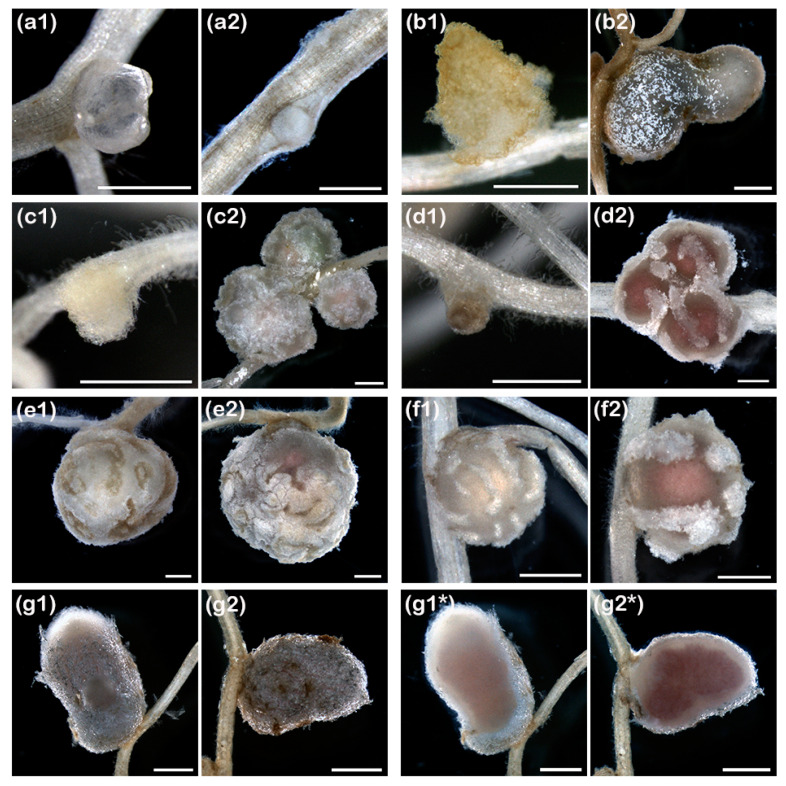
Representative examples of pseudo- or mature-like nodule structures formed by ANU265::pMiniSym2 (**1**) or NGR234 (**2**) on roots of, respectively: (**a**), *Stylosanthes guianensis* cv. Schofield; (**b**), *Cajanus cajan* cv. Lab22; (**c**), *Tephrosia vogelii*; (**d**), *Vigna radiata* cv. King; (**e**), *Lablab purpureus*; (**f**), *Vigna unguiculata* cv. Blackeye; and (**g**), *Leucaena leucocephala*. In (**g1***) and (**g2***), corresponding sections of the *L. leucocephala* nodules shown in (**g1**) and (**g2**), respectively. Scale bars in (**a1**), (**a2**), (**b1**) and (**c1**) are 0.5 mm. All other scale bars are 1 mm.

**Figure 3 genes-11-00521-f003:**
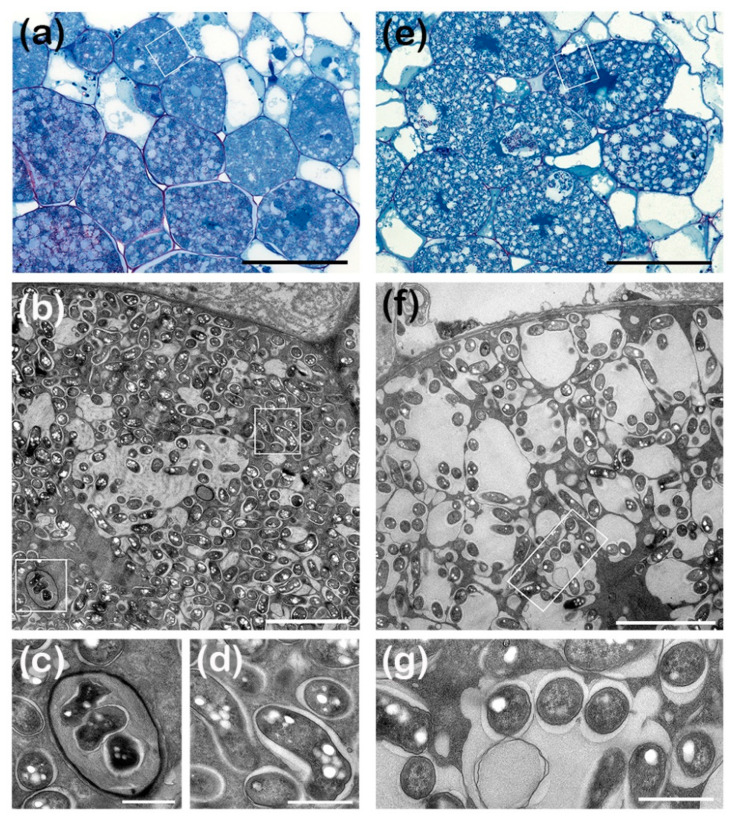
ANU265::pMiniSym2 formed persistent intracellular colonies in *L. leucocephala*. Micrographs of nodules harvested at 21 days post-infection with NGR234 in panels (**a**–**d**) and ANU265::pMiniSym2 in panels (**e**–**g**). Light micrographs (**a**,**e**) were of 500-nm thick sections whereas electron micrographs were of 85 nm thick sections of the same nodules. Sectors observed at higher magnifications are framed in white rectangles, with for example panels (**c**,**d**) corresponding to magnified areas of the nodule cell shown in (**b**). Higher magnification of NGR234 (**d**) and ANU265::pMiniSym2 (**g**) intracellular nodule bacteria. (**c**), cross section of one transcellular infection-thread. Poly*-β*-hydroxybutyrate (PHB) seen as electron-transparent droplets in bacteria. More than six nodules collected on several roots were processed per treatment. Scale bars are 50 µm in (**a**,**e**), 5 µm in (**b**,**f**) and 1 µm in panels (**c**,**d**,**g**).

**Table 1 genes-11-00521-t001:** Host-range of ANU265::pMiniSym2. Symbiotic phenotypes reported here correspond to the capacity of strains to form mature-like nodules (Nod+), pseudonodules (pNod) or no nodule at all (Nod-) after 28 days post-inoculation. Nod+/− refers to mature-like nodules formed on some of the inoculated roots. Legume species that were Nod- or pNod with ANU265::pMiniSym2 were assayed once. By contrast, at least seven consecutive nodulation assays were conducted on cowpea, *L. leucocephala* and siratro. Plants that were non-inoculated or inoculated with ANU265 never showed any nodule development. Phenotypes of NGR234 were consistent with those reported in [41,71]. *S. guianensis* cv. Schofield makes nodules of aeschynomenoid type (ATN). Previously thought to form indeterminate nodules (IDN), *C. cajan* and *T. vogelii* were recently reported to make determinate nodules (DN) with secondary clusters of dividing cells (DN*) [10]. Legume tribes are abbreviated as follows: Aeschynomeneae, Ae.; Crotalarieae, Co.; Cicereae, Ci.; Desmodieae, De.; Loteae, Lo.; Millettieae, Mil.; Mimoseae, Mim.; Phaseoleae, Ph.; Trifolieae, Tr.

Host Plant	Tribe	Nodule Type	ANU265:: pMiniSym2	NGR234
*Cajanus cajan* (Lab 22)	Ph.	DN*	pNod	Nod+
*C. cajan* cv. “Light Brown”	Ph.	DN*	pNod	Nod+
*Cicer arietinum* cv. Nayer	Ci.	IDN	Nod−	Nod−
*Crotalaria juncea*	Co.	IDN	Nod−	Nod+
*Desmodium intortum*	De.	DN	Nod−	Nod+
*Flemingia congesta*	Ph.	DN	Nod+/−	Nod+
*Glycine canescens 007*	Ph.	DN	Nod−	Nod+
*Lablab purpureus*	Ph.	DN	Nod+/−	Nod+
*Leucaena leucocephala*	Mim.	IDN	Nod+	Nod+
*Lotus corniculatus* cv. Malejovsky	Lo.	DN	Nod−	Nod−
*Lotus japonicus* cv. Gifu	Lo.	DN	Nod−	Nod+
*Macroptilium atropurpureum* cv. Siratro	Ph.	DN	Nod+	Nod+
*Medicago sativa* cv. Gemini	Tr.	IDN	Nod−	Nod−
*Medicago truncatula* cv. Jemalong	Tr.	IDN	Nod−	Nod−
*Mimosa pudica*	Mim.	IDN	Nod−	Nod−
*Pachyrhizus tuberosus*	Ph.	DN	Nod−	Nod+/−
*Stylosanthes guianensis* cv. Schofield	Ae.	ATN	pNod	pNod
*Tephrosia vogelii*	Mil.	DN*	pNod	Nod+
*Vigna radiata* cv King	Ph.	DN	pNod	Nod+
*Vigna unguiculata* cv. Blackeye	Ph.	DN	Nod+	Nod+
*V. unguiculata* cv. Kacang panjang	Ph.	DN	Nod+	Nod+
*V. unguiculata* cv. Red Caloona	Ph.	DN	Nod+	Nod+

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
