# Peer review of "A Minimal Genetic Passkey to Unlock Many Legume Doors to Root Nodulation by Rhizobia"

_genes, 2020, doi:10.3390/genes11050521_

Round 1

Reviewer 1 Report

The manuscript entitled “A minimal genetic passkey to unlock many legume doors to root nodulation by rhizobia” submitted to Genes is a study using synthetic biology to determine the minimal set of nodulation genes required to nodulate in a number of different legumes able to form indeterminate or determinate nodules.

I think the work described in the manuscript is of scientific merit and deserves publication in Gene. This opinion is supported by:

  • The low number of publications focused on the rhizobia-legume symbiosis in which techniques of Synthetic Biology has been used.
  • Rhizobia use combinations of different molecular signals to stablish successful symbiosis with different legumes. The importance of any particular signal will not be the same in all the legumes acting as compatible host. That means that the construction of plasmids containing combinations of an increasing number of different genes will be required in the future for acquiring further knowledge about the rhizobia-legume symbiotic specificity. Synthetic Biology appears as an efficient tool to obtain such chimeric plasmids.

The introduction provides the necessary information to understand the state of art concerning the specific topic investigated in the manuscript. In my opinion is concise and well written in terms of a proper organization of the information provided. I can not judge the level of the English language. It sounds very good to me, but it is only the personal opinion of a non-native English speaker.

Material and Methods describe properly the experiments carried out or provide a reference.

Apart of the molecular biology techniques used to construct the plasmids, most of the remaining experiments were mainly focused on nodulation tests using 17 different legumes. The nodules formed by Macroptilium atropurpureum, Vigna unguiculata and Leucaena leucocephala were investigated in detail by optic and electron microscopy. The authors chose these three legumes because nodules were formed upon inoculation with a NGR234 pSym-plasmid cured derivative (ANU265) carrying a plasmid (pMiniSym2) containing the NGR234 nodABCIJ, nodS, and nodD1 genes. In M. atropurpureum and V. unguiculata defence responses were detected. In Leucaena leucocephala, ANU265::pMiniPsym2 induced the formation of mature-like nodules that contained leghemoglobin and bacteria inside the nodules appears to survive.

The crisis provoked by the coronavirus makes very difficult to carry out complementary experiments. Thus, I will not ask for additional experiments. Nevertheless, I would suggest (not demand) just a few modifications/additions to the text:

  1. The manuscript does not describe any MS analyses to determine the chemical structure of the Nod Factors produced by ANU265::pMiniPsym2. I suppose that the chemical structure of the Nod Factors produced by ANU265::pMiniPsym2 is deduced from previous works on NGR234, its nodSU mutants, and by the Nod Factors produced by fredii USDA257 carrying the NGR234 nodSU genes (such as referenced 69 in the manuscript). I think it would be nice to make a Figure comparing the Nod Factors produced by NGR234 and those (real or deduced) of ANU265::pMiniPsym2. Do the authors know/suppose that the fatty acids of the Nod factors produced by these two strains are the same?
  2. juncea and G. canescens are not nodulated by ANU265::pMiniSym2. The authors state that “the keys missing for C. juncea and G. canescens to form visible module primordia may not be necessarily NF since T3SS effectors may either promote or abort nodule development”. The absence of nodules of Lotus japonicus and Desmodium intortum plants inoculated with ANU265::pMiniSym2 could not be included in this possibility? Could Pachyrhizus tuberosus also be included?

As I mention at the beginning of this report, I think that the manuscript deserves to be published in Gene. I think that this manuscript could the base of possible future works in which new genes will be added

Author Response

We thank Reviewer 1 for her/his considerate review and understanding of the current academic and sanitary situations, which precludes additional experiments in a realistic timeframe. As to reviewer 1 questions, our answers are as follow:

  1. Regarding the structure of nodulation factors (NF) synthesized by ANU265::pMiniSym2, it can be deduced from the many detailed chemical analyses carried out over many years on NF secreted by the parent strain NGR234. As explained in the manuscript, ANU265 is a derivative strain of NGR234 cured of its symbiotic plasmid pNGR234a. Thus, ANU265 carries none of the nodulation genes of pNGR234a that are responsible for NF synthesis. Genome sequence of the ANU265::pMiniSym2-Gus transconjugant (data not discussed in the manuscript) fully confirmed the identity of the strain and congruence with the NGR234 genome published by Schmeisser et al. (2009, Appl. Environ. Microbiol.) except, as expected, for the missing loci of pNGR234a. Thus, all of the following NF-modifying enzymes are absent from ANU265::pMiniSym2 : NodU (6-O-carbamoyltransferase), NodZ (6-O-fucosyltransferase), NoeE (sulfuryltransferase), NoeI (2-O-methyltransferase), NoeJ (mannose-1-phosphate ganuyltransferase), NoeK (phosphomanomutase), NoeL (GDP-mannose 4,6-dehydratase), NolK (fucose synthase), NolL (O-acetyltransferase) and NolO (3 or 4 O-carbamoyltransferase).

Since the function of each of these enzymes was demonstrated via biochemical analyses of NF secreted by mutant strains (for a summary please refer to the biosynthetic pathway detailed in Broughton et al. 2000 J. Bact.) we predicted that NF synthesised by ANU265::pMiniSym2 are methylated (NodS-dependent) pentamers of N-acetylglucosamine that should carry the same fatty acids (C18:1, C18:0 or C16:1) as those found on NF of NGR234. Following the suggestion made by Reviewer 1, we prepared a supplementary Figure S6 that shows the differences between NF of NGR234 (experimentally demonstrated) and those predicted for the ANU265::pMiniSym2 transconjugant. As Reviewer 4 considers the latter NF structure as purely theoretical, we don't know whether Figure S6 will be part of the published manuscript, however. Apparently, that choice will be in the hands of the scientific editor.

The second question asked by Reviewer 1, and comments made by Reviewer 2, prompted us to revisit the topic of the nodulation keys that are missing to ANU265::pMiniSym2 to make it nodulate a number of legumes that are hosts for NGR234. This is an extremely complex issue and in absence of additional experimental data, we hope the modifications we introduced in the second paragraph of the discussion will meet Reviewer 2 agreement.

Thus, we hope to have answered all of Reviewer 1 queries and would like, once again, to thank her/him for constructive comments and appreciating review.

Reviewer 2 Report

This manuscript investigates the basis for the unique promiscuity of the NGR234 rhizobial strain that nodulates an unusual and very broad range of fairly unrelated legume species. A mini plasmid, pMiniSym2, containing the core Nod factor biosynthesis genes is constructed and conjugated into a plasmid cured non-nodulating ANU265 derivative of the NGR234. This ANU265::pMiniSym2 transconjugant is then used to inoculate a range of legume species previously shown to nodulate effectively with NGR234 to determine which legume species respond to a Nod factor with a core structure  without additional decorations.  A subset of the tested legume species responded with nodule primordia or infected nodules while only Leucaena leucocephala develops leghemoglobin containing nodules.

The merit of results presented in this manuscript is to differentiate the Nod factor requirement for organogenesis and infection.  It is shown that the large cocktail of differently decorated Nod factor assumed to be responsible for the broad host range of NGR234 is not required to initiate nodule organogenesis on many of these host legumes while infection was impaired on most hosts. This result contributes to our understanding of NGR promiscuity.

Major comments:

There is at least one publication demonstrating bifurcation of the organogenesis and infection signal transduction pathways following Nod factor perception by the legume Nod factor receptors, see Madsen et al Ncomms 2010, doi: 10.1038/ncomms1009. This observation is very relevant for the results presented in the manuscript considering that receptor perception of different Nod factor structures may trigger the organogenesis but not infection pathway. This work should be discussed and referenced.

There is a higher copy number of the synthetic mini plasmid than the endogenous symbiotic plasmid pNGR234a. This could lead to a higher level of Nod factor biosynthesis in the ANU265::pMiniSym2 transconjugant which could by itself trigger the legume Nod factor receptors by mass action.  Ideally the level of Nod factor synthesized should be measured but the possible effects of increased synthesis should at least be discussed.

Author Response

We thank Reviewer 2 for her/his critical reading of our manuscript and for drawing our attention to the need of having a sustained infection process for continued nodule development. Accordingly, we have considerably revised the second last paragraph of the discussion to discuss the two early symbiotic pathways downstream of NF perception and in this context the possible role(s) of specific Nod-factor (NF) decorations in securing a robust infection of root tissue. We hope this changes will meet Reviewer 2 approval.

Regarding levels of NF secreted by the ANU265::pMiniSym2 and how increased synthesis may affect nodulation, this is an important and complex issue but we have little data to address it. So far, we have not compared the levels of NF produced by ANU265::pMiniSym2 and NGR234 in vitro. Even though analytical techniques have considerably improved since the 1990's when the first NF structures were reported, obtaining quantitative data on NF synthesis and secretion by rhizobia remains difficult. In vivo, that is on legume roots or in planta and on >20 hosts, this even more complex. Although we cannot exclude that ANU265::pMiniSym2 produces more NF than NGR234, we do not think this is the reason for the transconjugant relatively broad host-range for the following reasons: First, Cai et al. (2018, Plant Cell) convincingly showed that higher amounts of NF negatively affected the infection and nodulation of M. truncatula. Second, if increased levels of secreted NF were alone sufficient to trigger nodulation, some of the non-hosts (Nod-) for NGR234 could have responded positively to ANU265::pMiniSym2. Third, copy number of pMiniSym2 was measured and shown to be comparable to pNGR234a when cells were grown in minimal medium, and only slightly higher when a rich medium (TY) was used. Given these results we believe that in the rhizosphere, the copy numbers of symbiotic genes common to ANU265::pMiniSym2 and NGR234 are very close and if any difference existed it should not be necessarily a reason for having important differences in NF synthesis. Nevertheless, and following Reviewer 2 recommendation, we have addressed the question of NF levels in the second paragraph of the discussion.

Reviewer 3 Report

The paper touches an important problem of host specificity in rhizobia-legume symbiosys. Sinorhizobium fredii NGR234 is a very appropriate model in studying this matter. I find this research have perfectly set a task and succesfully solved it. 

Author Response

We thank Reviewer 3 for having accepted to review our manuscript and for her/his appreciation on the importance of the data reported herein.

Reviewer 4 Report

In the manuscript entitled “A minimal genetic passkey to unlock many legume doors to nodulation by rhizobia” Unay and Perret investigated the symbiotic promiscuity of Sinorhizobium fredii NGR234, a strain with a broad host spectrum. It has been long believed that the particularly large number of host plants that strain NGR234 nodulates is largely explained by the production of a large cocktail of different Nodulation factors. It has also been speculated that other features, like symbiotic effectors could also contribute to this striking phenotype. In this manuscript the authors challenge this hypothesis by exchanging NGR234s' natural symbiotic plasmid, which carries the nod factor synthesis genes and known symbiotic effectors by a synthetic plasmid that contains a minimal set of nod genes. These would theoretically lead to the production of a minimal Nod factor. They found that the strain carrying the minimal plasmid still could nodulate a large proportion of the tested hosts. However, in general it exhibited a reduced level of compatibility with the host plants at different stages of the symbiosis. For instance, it formed pseudonodules in Cajanus cajan and poorly infected nodules in Vigna unguiculata cv. Red Caloona and Macroptilium atropurpureum cv. Siratro. These are interesting observations, because they challenge a key and lock model, where one NF opens the door to one or few legumes. The paper is clearly written and the experiments are carefully documented with good images. However, there are some important controls that are missing to support some of the central conclusions of this manuscript. Also in some passages, statements are not supported by the data. Here I list some of the main points that should be revised by the authors.

Major comments:

1. In section 3.3 the authors test the nodulation range of strain ANU265::pMiniSym2 in comparison with the wild type strain NGR234. They use as negative controls non-inoculated plants and plant inoculated with the parental ANU265 strain. However the most adequate control would have been strain ANU265::pMiniSym1.

2. Lack of quantitative information in the material and methods. In general there is almost no information about the number of biological replicates and how many times experiments were independently conducted. This is important to conclude how representative are the results.

i) Section 2.3, Plant assays and analysis of root nodules. How many independent times was the experiment performed? How many plants were tested? How many nodules were analysed? How many sections?

ii) Section 2.4, Detection of plant immune responses. How many nodules were analysed? How many sections?

3. The phenotypic description is incomplete.

i) Qualitative and quantitate description of the shoot phenotype is missing. Were the plants nitrogen-starved or not? To address this provide colour of leaves, shoot pictures (if available), quantification of shoot dry weight.

ii) Quantification of the number of nodules. Table 1 only shows presence/absence of nodules. What are the criteria to define Nod+/Nod-? It is very important to know if Nod+ denotes that only one plant formed 1 nodule or all plants formed many nodules.

iii) Qualitative description of presence/absence of epidermal infection threads. This information is valuable, because Rodpothong and collaborators showed that lack of specific Nod factors substitutions impaired the formation of ITs in a host specific manner.

4. Claims not supported by the data.

i) Page 10, lines 360-362. "Together, these observations confirmed that ITs delivered ANU265::pMiniSym2 to nodule cells of V. unguiculata and M. atropurpureum, but that plant defences prevented the transconjugant from establishing persistent colonies inside of these nodule cells". No microscopic evidence of ITs is provided for Macroptilium atropurpureum cv. Siratro.

ii) Page 11, line 408-409. "Here, we showed that the situation is more complex, since methylated pentameric NF secreted by ANU265::pMiniSym2 act as a skeleton key that opens many legume doors". This nod factor is only theoretical. The authors did not conduct Nod factor structure determinations of strain ANU265::pMiniSym2. Unless this analysis is done, they should remove or re-write the statement.

iii) Page 12, line 444-445. "In this respect, that a strain making a single type of basic NF is capable of inducing sustained nodule formation..." As in point 4.ii. this has not been shown.

Minor comments:

1. Page 2, line 73-74. "Legumes decode NF mixtures via membrane bound LysM receptor kinases (LRK) that bind compatible NF with high affinity [27]" To my knowledge "decoding of NF mixtures" has not been yet shown. In the paper cited, only binding was shown.

2. Page 9, line 330. What do you mean by "circumscribed symbiosomes"?

3. Page 12, line 454. Production of leghemoglobin is not a good indication of nitrogen fixation. It it not necessarily correlated with the activity of the nitrogenase. The observations are not so surprising.

4. Page 12, line 458. What do you mean with pseudorhizobia?

5. Only use abbreviations when they are used multiple times. LRK is only used 3 times. It can be replaced in the text by "receptors".

6. How do your results relate to the work of Rodpothong et al 2009 MPMI? One of the main finding of your paper is that "the skeleton of the NF already unlocks the door to several legume roots...". Similar observations were made in this article.

7. Missing references.
i) Page 11, line 403. "...strains unable to form nodules". Check the rest of the text.

8. Page 3, lines 108-109. The authors write "pMiniSym2...allowed ANU265 to trigger nodule formation on roots of 10 of 15 legume species which associate with NGR234". This is somewhat misleading. Based on Table 1, NGR234 (15 nod+, 1 pnod, 1 nod+/-), ANU265::pMiniSym2 (7 nod+, 5 pnod).

9. Minor spelling inconsistencies: UK vs American English. E.g. colonize (line 41), colonisation (line 44). Check the rest of the text.

10. Figures and Tables.

i) Figure 3. Place all letters in the upper left corner.

ii) Figure S1. Instead of using A,B,C & 1,2, I recommend to label the species name on the side and the rhizobia strain on the top. This makes it easier to read the figure. I also noticed that depending of the program used to open the pdf, the picture will appear in colours or in black and white. I would recommend to ensure that the colour is not lost. This is crucial for the interpretation of the result.

iii) Figure S2. Idem as in 5.ii. The the roots in A and B are missing. Please indicate in the legend the direction of the cut.

iv) Figure S3. In the legend (A to D) all in bold. Can you indicate evidence for bacterial degradation in G? Why do you conclude that the structure shown on H is a remnant of an IT and not an abnormal IT structure or a different type of structure? Similar structures have been described in Lupinus, Aeschynomene and Lotus. Relate to that work (Gonzalez-Sama 2004 New Phytol; Bonaldi 2011 MPMI, Liang 2019 J Exp Bot)

v) Figure S4. A lower magnification would give a better overview of the cell structure.

Author Response

We thank Reviewer 4 for her/his extremely thorough reviewing of our manuscript and many critical but constructive remarks, which we tried to address in the revised document.

When appraising the revised versions of the manuscript and supplementary materials, please remind that the uploaded texts contain  modifications requested by all reviewers.

Major comments:

  1. Retrospectively, we agree with Reviewer 4 that using ANU265::pMiniSym1 instead of ANU265 would have been a better negative control. Given pMiniSym1 only carries four genes, none of which is even remotely connected to the symbiotic process, we failed to consider the plasmid itself as an appropriate control and apologize for it. Hopefully, having used ANU265 instead of ANU265::pMiniSym1 as a negative control will not be considered by Reviewer 4 as a reason to prevent publication.

2 and 3. To provide readers with additional experimental details (e.g. number of times nodulation assays were carried out, number of plants inoculated, number of nodules examined, etc.), these items are now listed in different sections of the manuscript where we think they are particularly relevant. We chose to include most of these details outside of the material and methods section because several of these items differed between plant species. We hope Reviewer 4 will be satisfied with these changes.

Regarding phenotypic descriptions, the main text now includes greater details and we added as supplementary material photographs of shoots and corresponding roots of cowpea, siratro and Leucaena leucocephala that were inoculated with NGR234, ANU265::pMiniSym2 or ANU265. These images clearly show that plants inoculated with ANU265 and ANU265::pMiniSym2 suffered from nitrogen starvation while those inoculated with NGR234 (or another proficient symbiont when NGR234 did not nodulate the targeted legume) appeared as fixing nitrogen. With these images, which are representative of the observed phenomena, we also hope to convince Reviewer 4 that the nodulation properties of the pMiniSym2 transconjugant are robust. Over the years, between 7 to 11 independent nodulation assays were carried out on these three plant species, with reproducible results. By contrast, most legumes that turned out to be Nod- or form pseudonodules when inoculated with ANU265::pMiniSym2 were tested once, with a minimum of 6 plants per inoculum.

We chose not to report nodule numbers because 28 days post-inoculation (dpi) is, in our experience and with our experimental system, a particularly early time point to secure reproducible results on nodule numbers. Because ANU265::pMiniSym2 does not carry any of the nif and fix genes and fails to fix atmospheric nitrogen in planta (see new Fig. S1), we do not think shoot dry weights as an essential dataset to the reported study. Because of Reviewer 4 constructive remark on consistence of reported nodulation phenotypes, we have now revised data for F. congesta and L. purpureus in Table 1. Since nodule formation was irregular on both plants, Nod+ was modified to Nod+/-, which is consistent with the similar nodulation behavior of NGR234 on P. tuberosus and because of an active T3SS (see Viprey et al. 1998 Mol: Microbiol.). All other Nod+ phenotypes reported in Table 1 are now consistent with observations made on all inoculated roots.

As to data on the presence/absence of epidermal infection threads, this is indeed a valuable and interesting information, but we feel it goes beyond the scope of the present manuscript. We agree that the early infection process is critical to symbiosis. Yet, providing a comparative, reliable and quantitative description of infection threads on all of the 16 hosts of Table 1 that are nodulated by NGR234 is not feasible given our current infrastructure, properties of wild type and transconjugant strains used in this work, diversity of legume responses and because distribution, numbers and size of root hairs vary considerably between these host plants. We plan to address this issue, but on a limited set of hosts, at a later stage once we will have secured appropriate reporter strains.

  1. Claims not supported by data.

(i) Regarding infection threads (IT) in siratro nodules, we do have several EM pictures showing structures resembling IT. Following the comments made by Reviewer 4 on IT in previous Figure S3 (now S4) and regarding various publications showing abnormal infection structures, we decided to modify the text and limit our claims to V. unguiculata.

(ii) As we did not determine the structure of ANU265::pMiniSym2 nodulation factors (NF), the statement was changed.

(iii) We have modified the conclusion section to insist upon the predictive nature of the NF structures synthesised by ANU265::pMiniSym2. In this respect, Reviewer 1 asked us to include a Figure comparing the structures of NGR234 and ANU265::pMiniSym2 NF, which has been added in the supplementary materials as Figure S6. We understand the concerns of Reviewer 4 on this issue. Nevertheless, NGR234 has been one of the few rhizobia for which enzymes modifying NF were thoroughly studied. Since ANU265::pMiniSym2 lacks functional NodU (6-O-carbamoyltransferase), NodZ (6-O-fucosyltransferase), NoeE (sulfuryltransferase), NoeI (2-O-methyltransferase), NoeJ (mannose-1-phosphate ganuyltransferase), NoeK (phosphomanomutase), NoeL (GDP-mannose 4,6-dehydratase), NolK (fucose synthase), NolL (O-acetyltransferase) and NolO (3 or 4 O-carbamoyltransferase) enzymes, the proposed NF structure is likely to be a bit more than just "theoretical". We have no particular feeling on Figure S6 but do agree with Reviewer 1 that it may help readers better assess the importance of "deleting" as many as 10 genes coding for NF modifying enzymes. We hope Reviewer 4 will not strongly oppose having such a Fig. S6 in the manuscript.

Minor comments:

  1. Introduction was changed to meet Reviewer 4 concern.
  2. "circumscribed symbiosomes" was replaced by "normal symbiosome structures".
  3. Leghemoglobin: sentence was modified.
  4. Define "pseudorhizobia": pseudo as not genuine. Sentence was modified to make it clearer.
  5. LRK abbreviations were kept only when necessary. Otherwise they were replaced.
  6. Likewise results reported in Rodpothong et al. (2009), our nodulation data suggests that some of the NF "decorations" may be essential for trigerring nodule organogenesis while other NF determinants may control early infection of roots. We have now considerably modified (including with the addition of new citations) the second paragraph before last of the discussion, to better address this issue. Although extremely important and essential to the subsequent seminal paper by Madsen and associates (2010), data presented in Rodpothong et al. refers to four legume species that belong to the same Lotus genus, however.
  7. References were added.
  8. In new Table 1: NGR234 (15 Nod+, 1 Nod+/-, 1 pNod) and ANU265::pMiniSym2 (5 Nod+, 2 Nod+/-, 5 pNod). Would Reviewer 4 agree with the perhaps less ambiguous statement of "…allowed ANU265 to trigger the nodule developmental program on roots of 10 of 15 legume species…"?
  9. Several spelling inconsistencies were corrected.
  10. Figures and Tables.

Lettering of all Figures was modified according to Genes style, and changes recommended by Reviewer 4 for Figures 3, old S1 (now S2) and old S3 (now S4) were made.

Regarding roots missing in now Figure S3, a short part of the root can be seen in the cowpea nodule infected with NGR234. For the ANU265::pMiniSym2 cowpea nodule, the root was close to the upper part of the image and cut transversal to the root axis. Fig. S3 legend has been modified accordingly.

Old Figure S4 (now S5). We feel a lower magnification would result in loosing details on the internal structure of the infected cells and we would prefer to keep it as it is.

We would like to thank again Reviewer 4 for having taken the time to carefully read our manuscript and also for having made so many constructive comments, some of them quite challenging. Hopefully, the revised text and Figures will satisfactorily address all her/his queries.

Round 2

Reviewer 4 Report

In the second version of the manuscript the authors improved the most important issues. I enjoyed reading the manuscript the first time and even more the second time. 

All the best